# Summertime Overheating Risk Assessment of a Flexible Plug-In Modular Unit in Luxembourg

**Michaël Rakotonjanahary** [1,*] **, Frank Scholzen** [1] **and Daniele Waldmann** [2]

1   Faculty of Science, Technology, and Medicine, Campus Kirchberg, University of Luxembourg, 1359 Luxembourg, Luxembourg; frank.scholzen@uni.lu
2   Faculty of Science, Technology, and Medicine, Campus Belval, University of Luxembourg, 4365 Esch-sur-Alzette, Luxembourg; daniele.waldmann@uni.lu
*   Correspondence: michael.rakotonjanahary@uni.lu

**Abstract:** Modular buildings offer faster construction process, provide better construction quality, allow reducing construction waste and are potentially flexible. Frames of modular units can be made of metal, timber, concrete or mixed materials but lightweight structures do not always allow erecting high-rise buildings and generally present a higher risk of overheating and/or overcooling. To reconcile these pros and cons, a typology of modular building called Slab was designed by a group of architects. The building is composed on the one hand of a permanent concrete structure named shelf-structure and on the other hand of several flexible removable timber modular units, also known as modules. The shelf-structure will host the common utility rooms and will serve as docking infrastructure for the housing modules. To provide high flexibility, the Slab building was designed to adapt to any orientation and location in Luxembourg. An energy concept and a HVAC systems design has been developed for the Slab building. Furthermore, a two-fold sustainability analysis was carried out. The first part of the analysis regards the determination of the minimum required wall thicknesses of the modules in accordance with Luxembourgish regulatory requirements, although the current regulation does not yet consider the Slab building typology. The second part, which is the subject of this paper, is thermal comfort assessment, more precisely, summertime overheating risk assessment of these modules, in compliance with Luxembourgish standard. In this regard, dynamic thermal simulations have been realized on two module variants; the first fulfills the passive house requirements, and the second—the current requirements for building permit application, which in principle corresponds to low energy house requirements. Simulations showed that with adequate solar shading and reinforced natural ventilation by window opening, overheating risk could be avoided for the normal residential use scenario for both module variants.

**Keywords:** plug-in architecture; modular building; flexible container unit; off-site construction; energy performance; dynamic thermal simulation; summertime overheating assessment

## 1. Introduction

From 1850–1900 to 2006–2015, the mean land surface air temperature increased by 1.53 °C, while the global mean surface (land and ocean) temperature increased by 0.87 °C [1]. According to scenarios developed by different organizations, global warming would get worse if no drastic measures are taken. The worst-case scenario Oceans developed by Shell company jointly with Massachusetts Institute of Technology, for instance, foresees a global average temperature rise of more than 2.5 °C from 1861–1880 to 2100 [2]. Another worst-case scenario proposed by the French National Centre for Scientific Research (CNRS) jointly with the French Alternative Energies and Atomic Energy Commission (CEA) and Météo-France forecasts an increase of the global average temperature of

6 °C up to 7 °C in 2100 [3]. To face the challenges of climate change, the European Union adopted in 2007 ambitious objectives for 2020. These were the reduction of greenhouse gas emissions by 20%, the increase of the share of renewable energy to 20% and the improvement by 20% in energy efficiency [4]. Buildings consume more than 40% of global energy and are responsible for one third of global greenhouse gas emissions [5]. Moreover, the lifecycle approach reveals that over 80% of greenhouse gas emissions take place during the operational phase of buildings (for heating, cooling, ventilation, lighting, appliances, and other applications) [5]. In this context, regulations regarding energy performance of buildings are subsequently more demanding. In the European Union, the Energy Performance of Buildings Directive (EPBD) 2010/31/EU [6], amended by the Directive 2018/844/EU [7], requires all new buildings to be nearly zero-energy buildings (NZEB) by the end of 2020. In this respect, building envelopes need to be well thermally insulated and designed in such a way as to take the most advantage of solar gains, without, as far as possible, exposing buildings to overheating risk. Moreover, buildings shall be constructed with eco-friendly materials and their heating, ventilation and air-conditioning (HVAC) systems shall be efficiently designed while using renewable energy sources as much as possible. Buildings shall ensure a minimum comfort to the tenants but on top of that, they could be expandable/shrinkable to be capable of adapting to the increase/decrease of the housing demand and be flexible so that they can fit any orientation and location. To combine all these criteria, modular constructions could be a solution.

A modular construction is an assembly of standardized-dimension building elements such as wall panel, slab, beam or also an assembly of container-type units called "modules," or else "prefabricated prefinished volumetric construction (PPVC)" which are prefabricated in factory and afterwards transported and assembled on-site. The naming of "container house" is given to transportable modules that are completely finished in the factory and ready to be inhabited; for smaller units, eventually with different shapes, the nomenclature of "living pod" or "capsule" is also found in literature. Modular buildings are always prefabricated buildings, but the reverse is not necessarily true. The degree of prefabrication and assembly technique mainly varies depending on the life span of the building (temporary or permanent), the desired space layout and the technical equipment to install (heating, electricity, sewer, plumbing, etc.). Compared to conventional constructions, modular buildings have remarkable advantages. Firstly, they offer a faster construction process [8–10] and provide an improved construction quality [10–13]. Secondly, they allow reducing construction waste [10,14,15], construction interruptions and nuisances generated on-site [10]. Lastly, they offer a great flexibility insofar as the modules can be refitted, relocated and refurbished. Frames of modules can be made of metal, timber [14], concrete [15] or mixed materials but lightweight structure do not always allow to erect high-rise buildings and generally present a higher risk of overheating and/or overcooling [16,17]. On the subject, Yoo et al. [18] carried out an interview on 23 residents of shipping container houses in Seoul and Gyeonggi-do (Korea). The survey indicated that 14 out of 23 were dissatisfied with the insulation of their housing. Regarding the environmental dimension, Aye et al. [19] conducted a study on a prefabricated case study building. It was found that the modular timber construction presents approximately the same embodied energy as the prefabricated concrete construction (10.49 GJ/m$^2$ and 9.64 GJ/m$^2$ of floor area, respectively), which is around 1.4 times less than that of the modular steel construction (14.40 GJ/m$^2$). The study also revealed that the total mass of the modular timber construction (0.25 t/m$^2$) is 4 times less than that of the modular concrete construction (1 t/m$^2$) and this aspect is very important for a construction to be portable or relocatable. Considering these pros and cons, a typology of modular building called Slab, as presented in Section 3, was designed by a group of architects and engineers within the team of the ECON4SD (Eco-Construction for Sustainable Development) research project. The core objective of the ECON4SD project is to develop new building components and design models to achieve maximum efficiency in terms of resource and energy use. The two architects of the ECON4SD project, M. Ferreira Silva and F. Hertweck, have furthermore developed two other building typologies, namely the Tower and the Demountable buildings; their

paper [20] provides more details on the design motivation and on the architectural aspect of these three building typologies.

The Slab building is a hybrid modular construction based on the plug-in concept or on Metabolism, from a wider perspective. The plug-in concept was imagined by Le Corbusier around the 1950s when he designed the Unité d'habitation (Cité Radieuse) [21] although it has never been realized in any of his buildings. The plug-in concept involves a primary structure, in which prefabricated housing units are slotted, whereas Metabolism [22] is a Japanese architectural movement established in the late 1950s, combining megastructures with the principles of biological growth, in order to allow buildings to expand/shrink. As a result, the Slab building is composed of a permanent reinforced concrete structure named shelf-structure, and several flexible removable timber modules used as housing units, as presented in Section 3. A two-fold study was carried out on the Slab building. The first part of the study is the development of an energy concept and a HVAC system design for the whole building. The second part involves a sustainability analysis on the modules, focusing on energy aspect and on thermal comfort. The sustainability analysis was realized in accordance with Luxembourgish regulatory requirements although the current regulation does not yet consider the Slab building typology. The energy analysis regards the determination of the wall thickness of a module, whereupon two module variants have been dimensioned. The first variant fulfills the requirements for AAA energy class, which correspond to the passive house requirements in Luxembourg and which is requisite to have a NZEB. The other one fulfills the current requirements for building permit application, which in principle corresponds to the low energy house requirements in Luxembourg. Since the two module variants have a relatively low thermal inertia as described in Section 4.2.1, thermal comfort is a sore spot, which deserves particular attention. Thus, this paper aims at assessing thermal comfort of the two module variants, more precisely, summertime overheating risk, which is also a key step allowing to check the necessity of an active cooling system. To appraise the impact of thermal mass on overheating, a fictive module version fulfilling the AAA energy class requirements but presenting a higher thermal inertia was studied.

## 2. Literature Review

Articles regarding thermal comfort assessment of modular constructions are scarce in literature. Fifield et al. [17] conducted a summertime overheating assessment of thermally lightweight, well-insulated, naturally ventilated modular healthcare buildings in the UK, according to British standards. Depending on the usage of the rooms, British standards propose two methods to assess overheating risk; these are modelling and in-use monitoring. The method used in their study was in-use monitoring, predicated on both static and adaptive overheating criteria. Static overheating criteria are only based on indoor environmental factors, more specifically on indoor temperatures (e.g., dry-resultant temperature, operative temperature, air temperature) in contrast with adaptive criteria, which additionally consider personal factors. The most popular thermal comfort assessment based on adaptive criteria is the American Society of Heating, Refrigerating and Air-Conditioning Engineers (ASHRAE) method. It proposes the predicted mean vote (PMV) model, under which the highest comfort temperature in winter and summer are 24.3 °C and 26.7 °C (new effective temperature), respectively [23]. Fifield et al. concluded that these modular buildings are at risk of summertime overheating in a relatively cool UK summer condition.

Regarding plug-in or flexible modular units, no studies on thermal comfort assessment have been found in literature; nonetheless, few ones about thermal analysis, which are closely related to thermal comfort assessment, have been listed. Ulloa et al. [24] realized a study on standard 20-foot equivalent unit (TEU) shipping containers made of COR-TEN®steel, reused as service modules (first aid module, shower module and refrigeration module) in the area of humanitarian help or social emergency. The purpose of the study was to estimate the peak for the heating and cooling demands in order to choose the adequate HVAC system for the modules. Modules were studied in five locations spread in different climate areas according to the Köppen–Geiger classification

(equatorial, arid, warm temperate, snow and polar climates) [25], whereas their orientation was fixed (window facing west). The thermal analysis involved dynamic thermal simulations (DTS) in TRNSYS software (v.17, Thermal Energy System Specialists, LLC, Madison, WI, USA) whereby modules geometry was modelled on Trimble SketchUp. The heating and cooling demands of the first aid module were predicated upon EN ISO 13790 [26] from the temperatures range which allows maintaining thermal comfort for a hospital that is from 22 °C to 26 °C. Kosir et al. [27] conducted a study to evaluate energy and visual (daylight) efficiency of a flexible prefabricated modular unit of 6.5 m length, 3.0 m width and 3.4 m height. The module was studied at five different locations (Reykjavik, Hamburg, Munich, Athens and Abu Dhabi) while varying different parameters, namely the orientation, the window-to-wall ratio (WWR), the window distribution, the shading, the thermal transmittance of the envelope and the glazing characteristics. The study required the realization of dynamic energy simulations on EnergyPlus software. The set-point temperatures for cooling activation were taken from the EN 15232 [28], which are 23 °C during the winter period and 26 °C during the summer period.

Buildings based on the plug-in concept are barely referenced in literature. Some of them are still in their pre-project phase, such as the "plug-in hexagonal housing units" [29] or the "plug-in modules system" [30], and others have been completed, such as the Nakagin Capsule Tower [31] and the NEST building [32]. Some Kasita living units [33,34] have also been achieved but the rack structure onto which they can be slotted still remains in project phase. No thermal analysis or thermal comfort assessment on these buildings has been found in literature except a post-occupancy evaluation on the Nagakin Capsule Tower. Indeed, two architects, L. Soares and F. Magalhães, lived in that building for almost a year and based on the remaining residents' observation, the indoor climate of the capsules is reported as too hot in summer and too cold in winter [35]. The two architects state that this is due to the fact that all wall surfaces of the capsules are in contact with the exterior. Nevertheless, the preponderant reason for that seasonal discomfort would be the low thermal inertia of the capsules' enclosure, knowing that capsules' walls are built with lightweight welded steel frames, filled with iron plates and insulated with asbestos [31].

## 3. The Slab Building

The Slab building has four open floors on the modules side and nine floors on the shafts side, as depicted in Figure 1c,d. The ground floor will host shops in urban locations and offices or workspaces in suburban areas. The top floor will be used as common space and the four open floors will accommodate the 48 plug-in modules. Each open floor can receive up to twelve stacked modules. The interior dimensions of the modules are 3 m width, 9 m length and 2.7 m height, as illustrated in Figure 1a,b; their wall thickness is 40 cm on all sides, as explained in Section 4.2.1. Given their size, the modules will be transported by special convoy since the maximum width allowed for standard road transport is set to 2.5 m [36] within the European Union. The building envelope of the modules will be built as much as possible with wood-based construction materials, seeing that these present a low embodied energy and are lightweight, as discussed in Section 1. The framing of the modules is made of timber I-beams connected to a timber column-beam structure, as shown in Figure 2c. Ducts and pipes for the HVAC system and for other technical equipment (electrical, sewer, plumbing, etc.) will be integrated into the roof/floor of the modules and will be connected/disconnected from the shelf-structure via a plug-in system, as shown in Figure 1b. A module offers 27 m$^2$ of living space but larger housing units can be realized by combining two or up to four units, as shown in Figure 1c. A concept for combining the modules in order to limit thermal bridging is being designed. Regarding the openings, a window of 3 m width and 2.7 m height is located on the front facade, as shown in Figure 2a. To maximize solar gains and daylight penetration, a non-operable window of 0.9 m width and 2.1 m height is located on the back facade, as presented in Figure 2b. A non-glazed door of 0.9 m width and 2.1 m height is also located on the back facade. The shelf-structure serves as docking space for the modules, provides building services including HVAC, ensures both vertical and horizontal circulation and hosts the common utility rooms. Thanks to a rail system, the modules can

be individually plugged/unplugged from the shelf-structure without affecting the adjacent modules. This allows the Slab building to extend or shrink and the modules to be relocated. These operations can be executed at any time with the help of a crane. The modules can be reused or recycled at the end of the first service life, depending on their condition and material degradation. If reused, they could be sent back to the manufacturing plant to be refurbished and eventually refitted. The Slab building is intended to be constructed in Luxembourg; therefore, the modules shall be designed to fit any orientation and location in Luxembourg to be flexible.

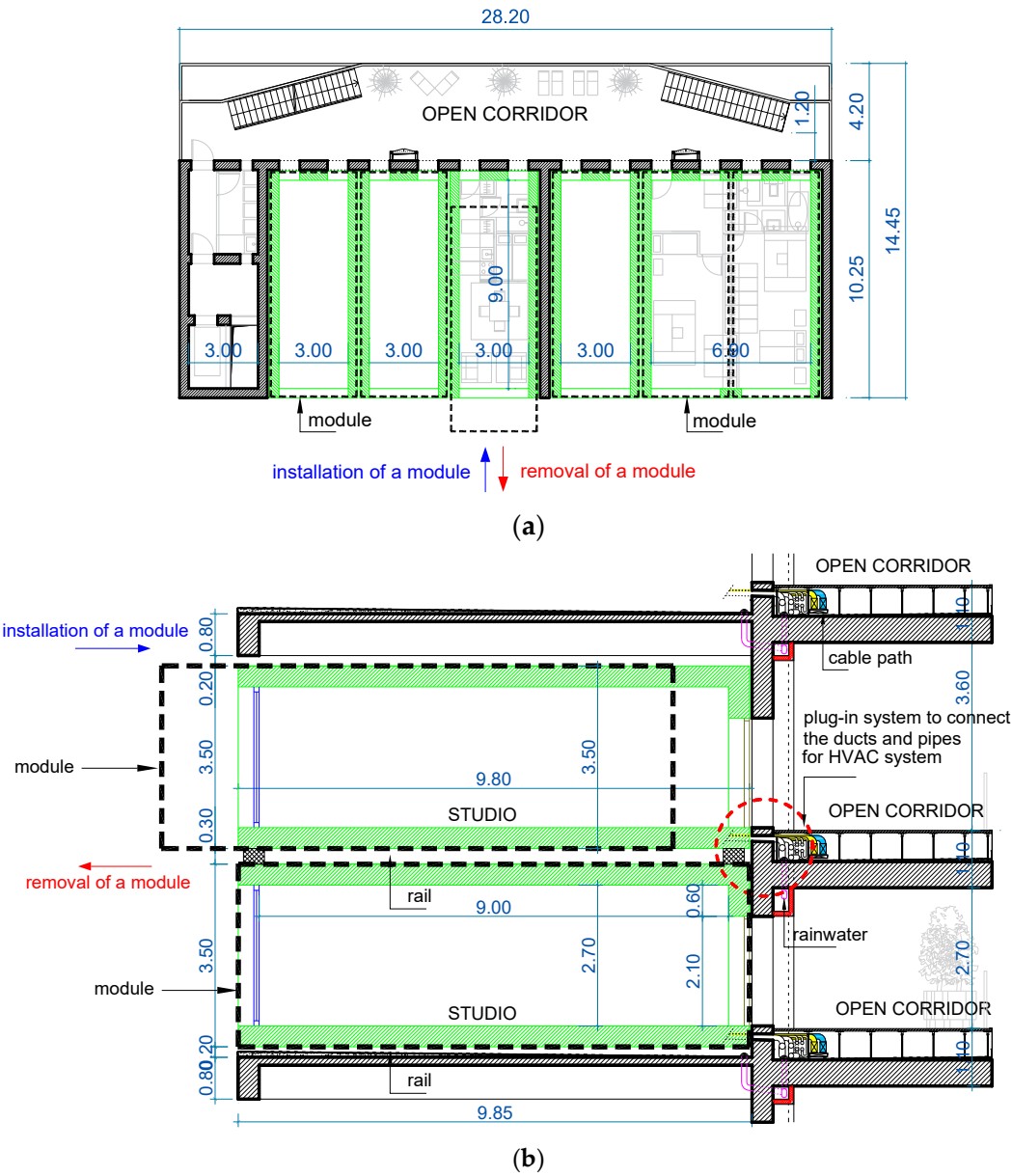

**Figure 1.** *Cont.*

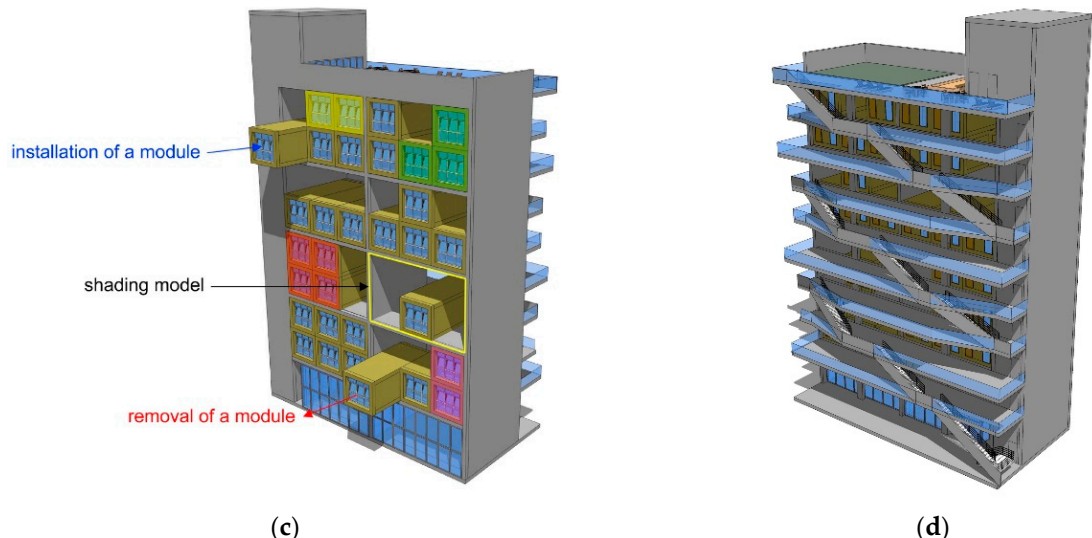

**Figure 1.** Drawings and 3D views of the Slab building: (**a**) Plan view of the current floor; (**b**) Cross section of the current floor; (**c**) 3D view of the front facade; (**d**) 3D view of the back facade.

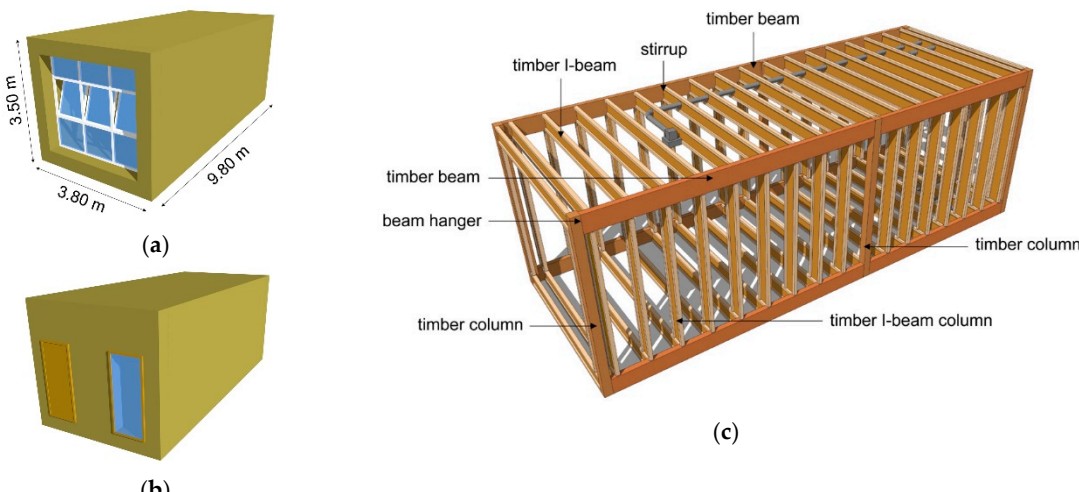

**Figure 2.** 3D views of the module: (**a**) Front facade; (**b**) Back facade; (**c**) Load-bearing structure.

## 4. Methods

This section is divided into three main parts. The first part explains the basis of the Luxembourgish regulation regarding overheating risk assessment on residential buildings. The second part presents the simulation model and the building model. The third part discusses the simulation parameters.

### 4.1. Overheating Risk Assessment According to Luxembourgish Regulation

On the basis of the DIN 4108-2, the Luxembourgish regulation on energy performance of residential buildings [37,38] prescribes two methods regarding summertime thermal comfort. The first one is the checking of the minimum requirements for summer thermal protection, and the second one is the assessment of summertime overheating risk, requiring the realization of DTS to check a static overheating criterion. The second method, which is used in this paper, involves a ratio of "overheating period" (OP) over "exploitation period" (EP) where the overheating period is the time during which the free-running ambient indoor temperature exceeds 26 °C. The exploitation period has not been further defined in the regulation; however, DIN 4108-2:2013-2 [39] defines an exploitation period of 24/7. Hence, if this ratio is below 10%, summertime thermal comfort is ensured. Regarding the

occupancy schedule, presented in Section 4.3.3, and the internal gains, described in Section 4.3.4, the Luxembourgish regulation does not dictate to refer to a specific standard.

### 4.2. Simulation Model and Building Model

DTS have been realized on TRNSYS software according to the simulation model shown in Figure 3.

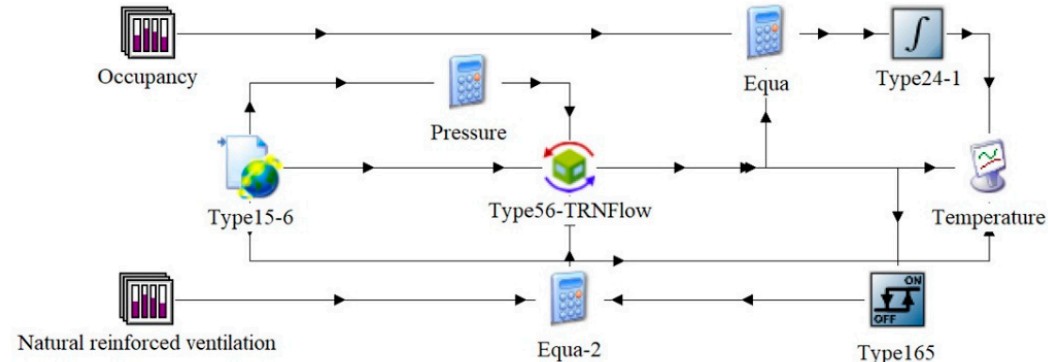

**Figure 3.** TRNSYS model for DTS.

The building model was created in TRNBuild and Sketchup software. The module model comprises a single thermal zone with residential building input characteristics. The module is assimilated to a single-family house implemented on a hosting site, which is none other than the shelf-structure. Module walls can be adjacent to other walls but to consider the worst-case layout, all walls of the module are assumed to be exposed to the outdoor influences. The building components of the shelf-structure create sun shading on the modules that has to be considered. In this regard, the shading/insolation matrix was generated by the 3D data geometry mode, which requires the 3D modelling of the shading. The shading model denoted in Figure 1c represents the worst-case sun shading on a module since this configuration exposes the most walls of the module to the sun; it corresponds to half of the current floor. Note that a shear wall is splitting the current floor into two parts. The 3D sun shading model of the shelf-structure is illustrated in Figure 4. The surfaces on the top, the sides and the back facade of the model represent the shading generated by the slab, the walls and the open corridors, respectively. As the width of the open corridors is variable, the smallest width (1.20 m) was taken.

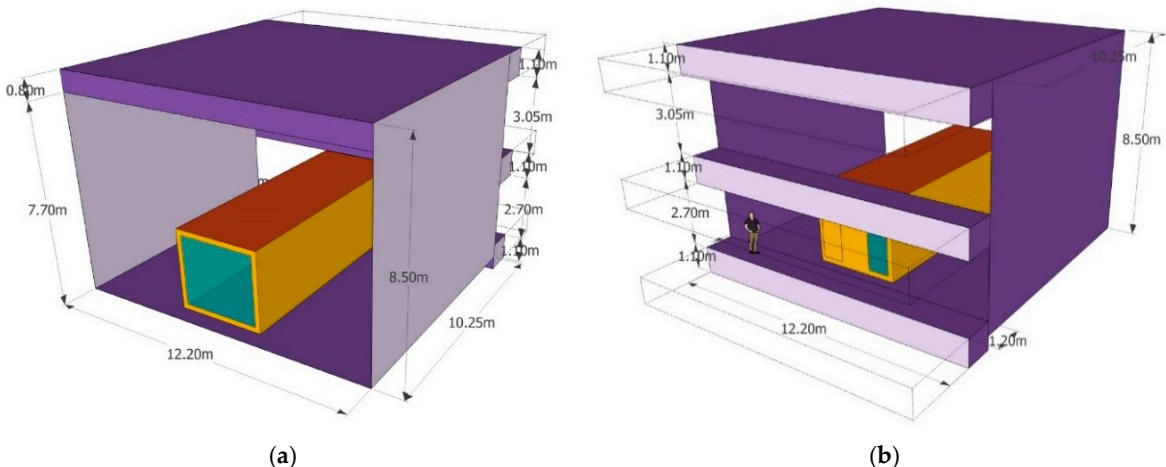

| (a) | (b) |
|-----|-----|

**Figure 4.** 3D sun shading model of the shelf-structure on the module: (**a**) Front facade; (**b**) Back facade.

The minimum required wall thicknesses were determined in accordance with the Luxembourgish regulation on energy performance of residential buildings and based on the worst-case orientation

(window facing north), which makes the module variants very well thermally insulated. For the current requirements for building permit application, characteristics of walls, windows and door are similar to those found in typical low energy houses. For the requirements for AAA energy class, characteristics of these components were set to very high performance.

### 4.2.1. Opaque Walls on the Module Envelope

Basically, the six sides of the module envelope have the same structure. However, additional elements could eventually be added, such as a heating floor system, for instance. Energy balance calculations have shown that a wall thickness of 40 cm is sufficient to fulfil the two requirements explained in Section 4.2. For the AAA energy class requirements, 31 cm of aerogel is required, resulting in a wall U-value of 0.062 W/m$^2$.K. For the current requirements for building permit application, 31 cm of wood wool is sufficient, giving a wall U-value of 0.123 W/m$^2$.K. An additional layer of 5 cm lightweight concrete was applied on the walls and the floor of the fictive module version, fulfilling the AAA energy requirements, and this is the only difference between the fictive and the current versions; hence, wall U-value of the fictive module version remains at 0.062 W/m$^2$.K. The structure of the module walls for both variants is illustrated in Figure 5.

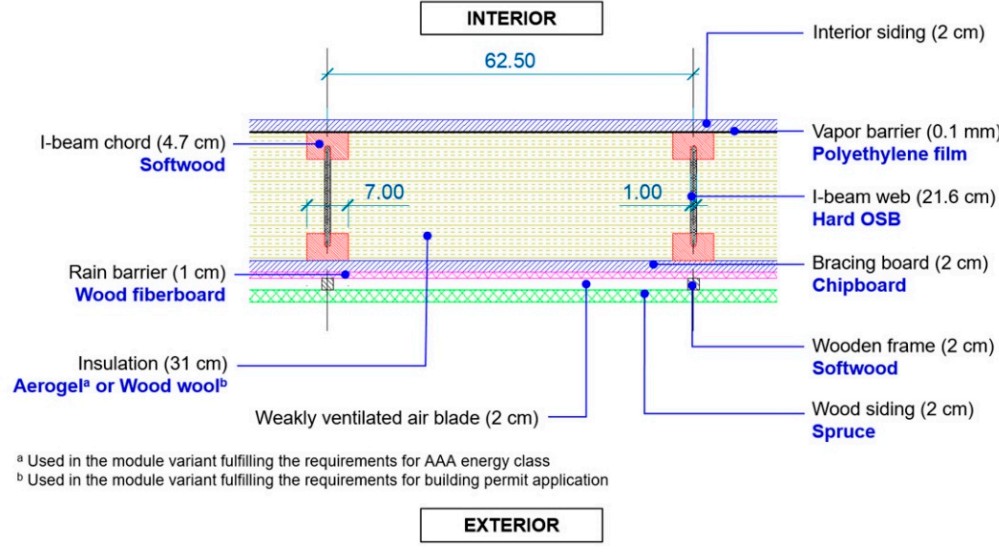

**Figure 5.** Structure of the module walls.

Characteristics of materials on the opaque walls of the module envelope are given in Table 1.

**Table 1.** Characteristics of materials on the opaque walls of the module envelope.

| Building Material | Heat Conductivity W/(m.K) | Heat Capacity Wh/(kg.K) | Density (kg/m$^3$) |
|---|---|---|---|
| Chipboard * | 0.14 | 0.47 | 500 |
| Aerogel [a] [40] | 0.02 | 0.28 | 150 |
| Wood wool [b] [41] | 0.04 | 0.58 | 50 |
| Wood fiberboard * | 0.09 | 0.69 | 650 |
| Air blade [c,*] | 0.03 | 0.28 | 1.23 |
| Hardwood * | 0.18 | 0.44 | 700 |
| Softwood * | 0.14 | 0.61 | 450 |
| Lightweight concrete [d,*] | 1.80 | 0.28 | 1400 |

[a] Used in the module variant fulfilling the requirements for AAA energy class. [b] Used in the module variant fulfilling the requirements for building permit application. [c] The value of the heat conductivity varies according to the thickness and the exchange with external. [d] Used in the fictive module version of the variant fulfilling the requirements for AAA energy class. * Source of the values: Lesosai software database.

Table 1 indicates that both module variants have a relatively low thermal inertia but the module variant fulfilling the requirements for building permit application has a slightly higher thermal mass.

### 4.2.2. Transparent Walls on the Module Envelope

Characteristics of the transparent walls of the module envelope are given in Table 2.

**Table 2.** Characteristics of the transparent walls on the module envelope.

| Components | Module Variant | |
|---|---|---|
| | **Building Permit Application** | **AAA Energy Class** |
| **Glazing and frame:** | | |
| $U_{glazing}$ | 1.10 W/(m$^2$.K) | 0.55 W/(m$^2$.K) |
| g-value | 60% | 60% |
| $U_{frame}$ | 1.10 W/(m$^2$.K) | 0.70 W/(m$^2$.K) |
| **Non-operable window:** | | |
| Gross dimensions (width × height) | 0.90 m × 2.10 m | 0.90 m × 2.10 m |
| Surface ratio glazing/window | 70% | 80% |
| $U_{installed}$ | 1.21 W/(m$^2$.K) | 0.65 W/(m$^2$.K) |
| **Window** | | |
| Gross dimensions (width × height) | 3.00 m × 2.70 m | 3.00 m × 2.70 m |
| Opening | Tilt and turn window | Tilt and turn window |
| Gross dimensions of the operable part (width × height) | 3.00 m × 1.20 m | 3.00 m × 1.20 m |
| Surface ratio glazing/window | 75% | 80% |
| $U_{installed}$ | 1.20 W/(m$^2$.K) | 0.64 W/(m$^2$.K) |

*4.3. Simulation Parameters*

4.3.1. Simulation Period and Weather Data File

The selected simulation period is from the 1 of June to the 31 of August, accounting for 2184 h. Luxembourg is situated in Western Europe, between Belgium, France and Germany. The country is divided into two regions: the Oesling in the north (225–559 m above sea level) and the Gutland in the south (133–440 m above sea level). Despite the altitude difference between the two regions, Luxembourg climate is relatively homogeneous, and is classified as Cfb, i.e., warm temperate climate with fully humid precipitation and warm summer temperature according to the Köppen–Geiger world climate map [25]. The weather data file "DE-Trier-106090.tm2" was used, given that Luxembourg has similar weather conditions to the German city Trier (latitude 49.75° N, longitude 6.67° E), which is just 15 km from the national border. The altitude of the weather station and the Slab building are 278 m and 133 m (lowest altitude in Luxembourg), respectively. The curves of temperature and relative humidity are presented in Figure 6, the curves of wind direction and wind speed—Figure 7, the curves of global horizontal radiation and direct normal radiation—Figure 8. The values of the global horizontal radiation and the direct normal radiation are not interpolated.

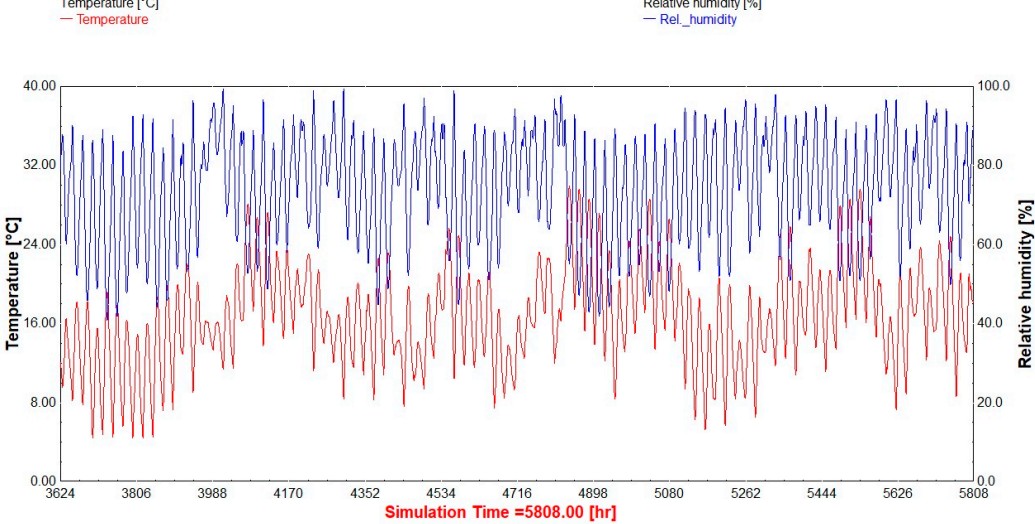

**Figure 6.** Temperature and relative humidity.

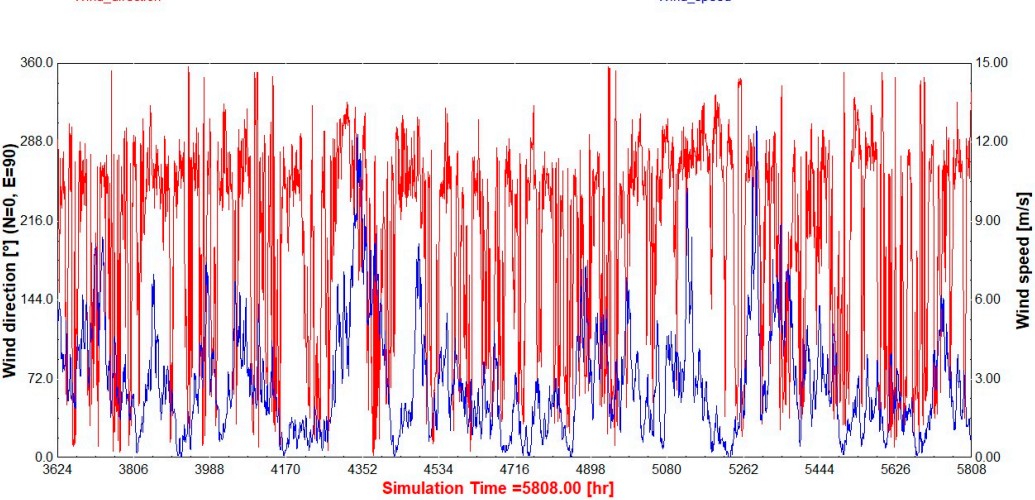

**Figure 7.** Wind direction and wind speed.

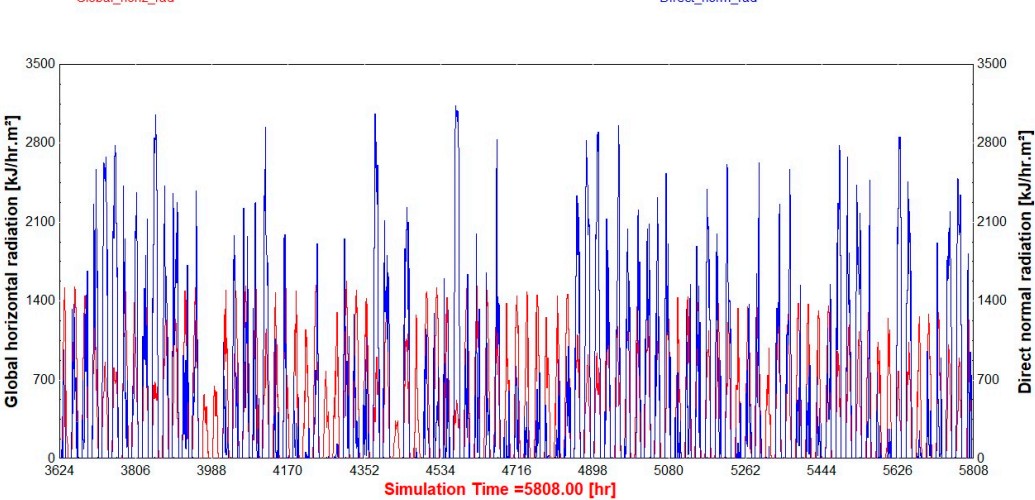

**Figure 8.** Global horizontal radiation and direct normal radiation.

### 4.3.2. Occupancy Scenarios and Mechanical Ventilation Airflow

A high and a low occupancy scenario were studied to consider possible worst-case scenarios. For the high occupancy scenario, the studio is occupied by two persons at all times, whereas for the low occupancy scenario, it is unoccupied from 9 a.m. to 6 p.m. on weekdays as well as on weekends. When the room is occupied, the mechanical ventilation provides an airflow of 60 m³/h, based on an airflow per person of 30 m³/h. When the room is unoccupied, the mechanical ventilation provides a minimum hygienic air change rate of 0.35 h$^{-1}$ corresponding to an airflow of 23.6 m³/h, as required by Luxembourgish regulation. For the two occupancy scenarios, the exploitation period is 2184 h.

### 4.3.3. Internal Gains

The CIBSE TM59:2017 [42] was used as reference in this work as it proposes internal gain values applicable to studios following a well-defined profile. In CIBSE TM59:2017, the occupancy to consider corresponds to two people occupying the studio at all times on weekdays as well as on weekends. The total peak load from people is 150 W of sensible heat and 110 W of latent heat. Lighting is switched on from 6 p.m. to 11 p.m. with a load value of 2 W/m² of net floor area, thus, 54 W for the studio of 27 m². The equipment peak load is assumed from 6 p.m. to 8 p.m. with a value of 450 W, 200 W from 8 p.m. to 10 p.m., 110 W from 9 a.m. to 6 p.m. and from 10 p.m. to 12 p.m., and a base load of 85 W for the rest of the day. No equipment load is considered when the room is unoccupied. The internal gains profile on a daily basis for the two occupancy scenarios is illustrated in Figure 9. In the summer period, the heating system will be turned off and the supply air will be the outside air without any handling.

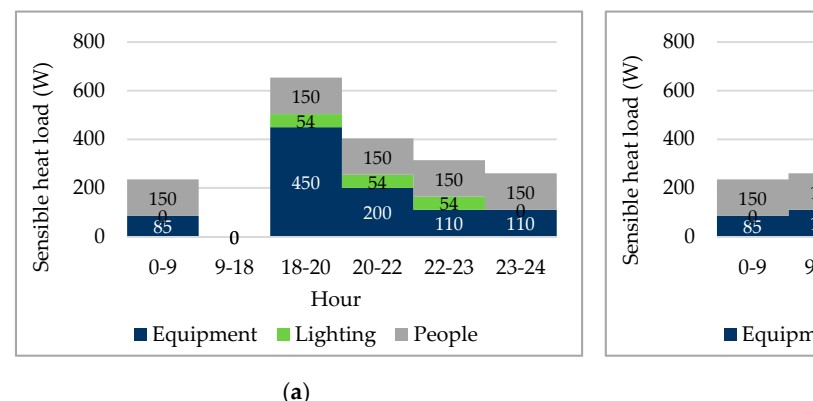
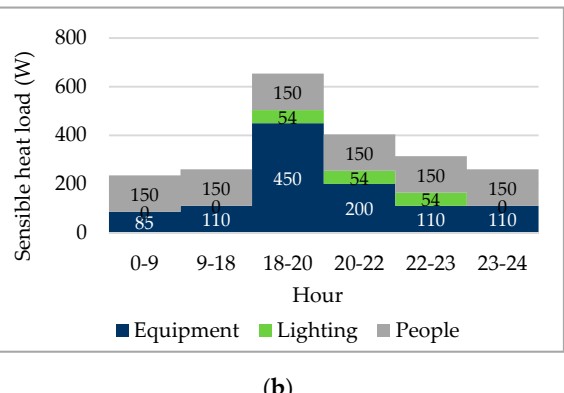

(**a**)　　　　　　　　　　　　　　　　　　(**b**)

**Figure 9.** Internal gains profile on a daily basis: (**a**) Low occupancy scenario; (**b**) High occupancy scenario.

### 4.3.4. Sub-Variants

Three sub-variants, S1, S2 and S3, defined according to different shading device configurations, were set up for each module variant mentioned previously in Section 1. S1 involves no shading device, S2—a fixed external shading device and S3—a moveable external shading device. The external shading device was chosen for its higher effectiveness in comparison to the internal since the material and the dimensions are identical [43]. An external shading device will be implemented on all transparent walls. The S2 sub-variant can also be assimilated to the case where the user does not adjust the shading device on purpose or by omission. For this sub-variant, the shading factor was arbitrarily set to a fixed value of 50% at all times as not to compromise daylight penetration. For the S3 sub-variant, the control of the external shading device is automated, according to the global radiation on window, i.e., on the facade. The upper radiation threshold values found in literature depend on the orientation and climatic conditions [44]. Newsham [45] conducted a study on the implication of manual control of window blinds on comfort and energy consumption for the climate of Toronto (Canada). He identified that the radiation value at which most users operated the blinds was 233 W/m². Lee et al. [46] suggested a value of 200 W/m² in a study about the evaluation of thermal and lighting energy performance of

shading devices on kinetic facades for the climate of Dubai (UAE). Wankanapon [47] found a value of 189 W/m² for a white roller shade allowing to save cooling energy from 21 to 27% for the climate of Minneapolis (USA). Therefore, the default values of radiation threshold set in TRNBuild were chosen for the simulations, whereby the shading device closes above 180 W/m² and opens bellow 160 W/m². For each sub-variant, three different ventilation scenarios are proposed. The first one is without natural reinforced ventilation, the second one with night natural ventilation from 10 p.m. to 7 a.m. and the third one with day and night natural reinforced ventilation. Natural reinforced ventilation will be operated automatically by window opening; the window closes when the room temperature drops to 19 °C and opens at over 22 °C.

### 4.3.5. Air Exchange Rates

Since the lowest atmospheric pressure is found at the lowest altitude, the module located at the lowest level (worst-case) was chosen for the simulation. The air exchange rates were simulated on TRNFlow according to the network model illustrated in Figure 10a. The "heights of link" of the different airflow network components on the Slab building are illustrated in Figure 10b.

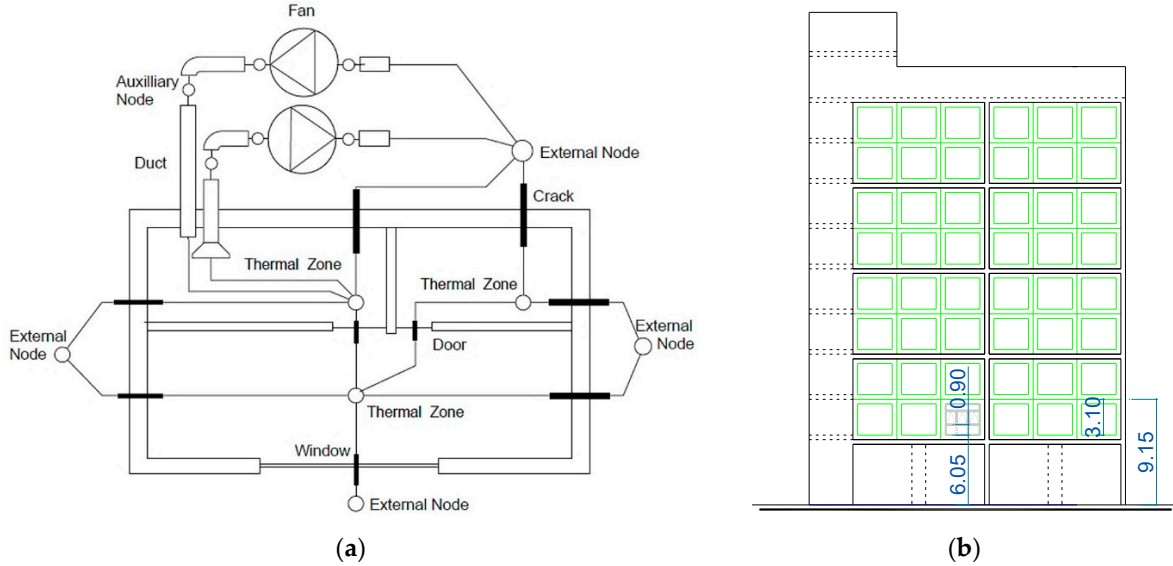

(**a**)　　　　　　　　　　　　　　　　　　　(**b**)

**Figure 10.** Model and drawing used on TRNFlow: (**a**) Network model on TRNFlow [48]; (**b**) Heights of the different airflow components on the Slab building.

No auxiliary node or ducts were modeled as the supply and extract airflow were assumed to be independent of the pressure difference. A tilt opening was considered for the window. If the maximum opening corresponds to an opening factor of 1, a value of 0.25 is chosen to avoid eventual high air velocity on natural ventilation. No natural reinforced ventilation is operating when the room is unoccupied. Regarding the air infiltration through the module envelope, Luxembourgish standard prescribes exchange rates at 50 Pa (n50 or ACH50) of 0.6 h⁻¹ and 1 h⁻¹ for the module variant fulfilling the AAA energy class requirements and the one fulfilling the building permit application requirements, respectively. To simplify, the air infiltration is assumed to occur only through the cracks around the windows and the door. In TRNFlow, the air exchange rates have to be entered in kg/s at 1 Pa and for that, the power law model of airflow through orifice can be used to estimate the airflow at different pressures, since there is a linear correlation between pressure difference and airflow. The power law equation of airflow through orifice is given by the following formula [49,50]:

$$Q = C * \Delta P^n, \tag{1}$$

where $Q$ is the airflow expressed in m³/s, $C$ is the air leakage coefficient, $\Delta P$ is the pressure difference and $n$ is the pressure exponent. Most cracks have a mixed flow regime with a flow exponent of 0.6 to 0.7 [48], therefore, a value of 0.65 was chosen. An air density of 1.2 kg/m³ was considered to determine the air mass flow. Table 3 presents the determination of the air mass flow at 1 Pa (Q1).

**Table 3.** Determination of the airflow at 1 Pa (Q1).

| Module Variants | n | V (m³) | ACH50 (h⁻¹) | Q50 (m³/h) | C | Q1 (m³/h) | Q1 (kg/s) |
|---|---|---|---|---|---|---|---|
| AAA energy class | 0.65 | 72.9 | 0.6 | 43.74 | 3.43 | 3.43 | $11.4 \times 10^{-4}$ |
| Building permit application | | | 1 | 72.90 | 5.73 | 5.73 | $19.1 \times 10^{-4}$ |

The air mass flow coefficient Cs is assumed to be commensurate with the crack length. Table 4 presents the determination of the air mass flow coefficient Cs at 1 Pa for each crack.

**Table 4.** Determination of the air mass flow coefficient Cs at 1 Pa for each crack.

| Cracks around the Element | Length (m) | Air Mass Flow Coefficient Cs (kg/s) | |
|---|---|---|---|
| | | "AAA Energy Class" Module | "Building Permit Application" Module |
| Door | 6 (26%) | $2.9 \times 10^{-4}$ | $4.9 \times 10^{-4}$ |
| Non-operable window | 6 (26%) | $2.9 \times 10^{-4}$ | $4.9 \times 10^{-4}$ |
| Window | 11.4 (48%) | $5.6 \times 10^{-4}$ | $9.3 \times 10^{-4}$ |
| Total | 23.4 (100%) | $11.4 \times 10^{-4}$ | $19.1 \times 10^{-4}$ |

The wind pressure acting on the facades was taken into account and the wind pressure coefficients Cp were defined, as shown in Table 5, section External nodes. Data and parameters entered on TRNFlow are also presented in Table 5.

**Table 5.** Data and parameters entered on TRNFlow.

| Parameters | Values |
|---|---|
| **Dual flow ventilation system** | |
| ▪ Supply/extract airflow (imposed) | |
|   -  Room unoccupied | 0.35 [1/h] = 23.6 m³/h |
|   -  Room occupied | 60 m³/h |
| ▪ Air density at Test Conditions | 1.2 kg/m³ |
| ▪ Air Mass Flow Coefficient Cs (if fan is turned off) | 0.2 kg/s at 1 Pa |
| ▪ Air Flow Exponent n (if fan is turned off) | 0.65 [a] |
| ▪ Height of link relative to "From-Node" | 9.15 m |
| ▪ Height of link relative to "To-Node" | 3.10 m |
| **Thermal airnode** | |
| ▪ Reference height | 6.05 m |
| ▪ Airnode interior dimensions: height / depth | 2.70 m/9.00 m |

**Table 5.** *Cont.*

| Parameters | Values |
| --- | --- |
| **External nodes** | |
| ▪ Reference height of Cp-values | 9.15 m |
| ▪ Wind direction angle: | |
| Length-to-width ratio: 2:1 | |
| Shielded (worst-case based on DTS) | |
| - EN001: wind from south-west (front façade): | $0°=-0.32$; $45°=-0.3$; $90°=0.15$; $135°=0.18$; $180°=0.15$; $225°=-0.3$; $270°=-0.32$; $315°=-0.2$ [51] |
| - EN002: Wind from north-east (back façade): | $0°=0.15$; $45°=-0.3$; $90°=-0.32$; $135°=-0.2$; $180°=-0.32$; $225°=-0.3$; $270°=0.15$; $315°=0.18$ [51] |
| **Crack around the door/non-operable window** | |
| ▪ Air Mass Flow Coefficient Cs | $2.9 \times 10^{-4}$ kg/s at 1 Pa ("AAA energy class" module) |
| | $4.9 \times 10^{-4}$ kg/s at 1 Pa ("Building permit application" module) |
| ▪ Air Flow Exponent n | 0.65 [a] |
| ▪ Height of link relative to "From-Node" | 6.05 m |
| ▪ Height of link relative to "To-Node" | 0 m |
| ▪ Connected to external node | EN002 |
| **Crack around the window** | |
| ▪ Air Mass Flow Coefficient Cs | $5.6 \times 10^{-4}$ kg/s at 1 Pa ("AAA energy class" module) |
| | $9.3 \times 10^{-4}$ kg/s at 1 Pa ("Building permit application" module) |
| ▪ Air Flow Exponent n | 0.65 [a] |
| ▪ Height of link relative to "From-Node" | 6.05 m |
| ▪ Height of link relative to "To-Node" | 0 m |
| ▪ Connected to external node | EN001 |
| **Window opening** | |
| ▪ Own height factor | 1 (window is in a vertical wall) |
| ▪ Category of opening | Bottom hinged sash window/door |
| ▪ Max. width/height of opening | 2.75 m / 1.05 m |
| ▪ Height of pivoting axis (A-height) | 0.90 m |
| ▪ Discharge coefficient Cd1 [b] (completely closed) | 0.6 |
| ▪ Discharge coefficient Cd2 [b] (completely opened) | 0.6 |
| ▪ For Closed Opening: | |
| - Flow coefficient Cs per m crack length | 0 kg/s/m at 1 Pa as described in Section 4.3.5 |
| - Flow exponent n | 0.65 [a] |
| ▪ Opening factor of window | maximum value of 0.25 |
| ▪ Connected to external node | EN001 |
| ▪ Height of link relative to "From-Node" | 6.95 m |
| ▪ Height of link relative to "To-Node" | 0.90 m |
| **Wind velocity profile** | |
| ▪ Wind angle direction | 0° |
| ▪ Wind velocity exponent of building location | 0.25 (wood, small city, suburb) |

[a] Most cracks have a mixed flow regime with a flow exponent n of 0.6 to 0.7 so the default value of 0.65 is taken. [b] For usual situations, a Cd value of 0.6 to 0.7 can be found very often in literature, so the default value of 0.6 is taken.

## 5. Results and Discussion

DTS showed that the worst-case orientation is window facing south-west, which was taken as reference for the results presented below. Table 6 presents the overheating periods of the two module variants for the low and the high occupancy scenarios according to two overheating criteria (>28 °C and >26 °C).

**Table 6.** Overheating periods of the two module variants for the low and high occupancy according to different overheating criteria.

| Module Variants and Sub-Variants | Low Occupancy | | | | | | | High Occupancy | | | | | | |
| --- | --- | --- | --- | --- | --- | --- | --- | --- | --- | --- | --- | --- | --- | --- |
| | Overheating Criteria | | | | Max. Room Temp. [°C] | ACH * [h⁻¹] | | Overheating Criteria | | | | Max. Room Temp. [°C] | ACH * [h⁻¹] | |
| | >28 °C | | >26 °C | | | | | >28 °C | | >26 °C | | | | |
| | OP [hrs.] | OP [hrs.] | OP/EP [%] | ODH [°Ch] | | Mean | Max | OP [hrs.] | OP [hrs.] | OP/EP [%] | ODH [°Ch] | | Mean | Max |
| **"AAA energy class" module** | | | | | | | | | | | | | | |
| **S1: Without any external shading device** | | | | | | | | | | | | | | |
| - without natural reinforced ventilation | 2135 | 2148 | 98 | 22,667 | 52 | 0.68 | 0.96 | 2142 | 2151 | 98 | 23,745 | 52 | 0.88 | 0.96 |
| - with night natural reinforced ventilation | 810 | 1146 | 52 | 4732 | 44 | 1.87 | 5.20 | 954 | 1285 | 59 | 5987 | 46 | 2.12 | 5.25 |
| - with day and night natural reinforced ventilation | 496 | 816 | 37 | 2450 | 40 | 2.38 | 5.07 | 351 | 649 | 30 | 1716 | 38 | 3.41 | 4.70 |
| **S2: With a fixed external shading device** | | | | | | | | | | | | | | |
| - without natural reinforced ventilation | 1484 | 1927 | 88 | 7787 | 40 | 0.67 | 0.95 | 1781 | 2081 | 95 | 10,810 | 40 | 0.86 | 0.95 |
| - with night natural reinforced ventilation | 218 | 505 | 23 | 872 | 35 | 1.60 | 4.86 | 424 | 792 | 36 | 1817 | 37 | 1.89 | 4.85 |
| - with day and night natural reinforced ventilation | 113 | 284 | 13 | 422 | 33 | 1.98 | 4.47 | 162 | 337 | 15 | 608 | 34 | 2.87 | 4.53 |
| **S3: With a moveable external shading device** | | | | | | | | | | | | | | |
| - without natural reinforced ventilation | 678 | 1249 | 57 | 2487 | 33 | 0.66 | 0.95 | 1233 | 1723 | 79 | 5348 | 35 | 0.86 | 0.95 |
| - with night natural reinforced ventilation | 28 | 151 | 7 | 105 | 30 | 1.42 | 4.38 | 159 | 416 | 19 | 546 | 32 | 1.75 | 4.60 |
| - with day and night natural reinforced ventilation | 8 | 79 | 4 | 35 | 29 | 1.73 | 4.30 | 45 | 188 | 9 | 163 | 31 | 2.69 | 4.40 |
| **"Building permit application" module** | | | | | | | | | | | | | | |
| **S1: Without any external shading device** | | | | | | | | | | | | | | |
| - without natural reinforced ventilation | 1464 | 1788 | 82 | 9457 | 46 | 0.68 | 1.03 | 1611 | 1915 | 88 | 11,134 | 47 | 0.87 | 1.03 |
| - with night natural reinforced ventilation | 601 | 896 | 41 | 3115 | 42 | 1.68 | 4.99 | 737 | 1044 | 48 | 4103 | 44 | 1.93 | 5.05 |
| - with day and night natural reinforced ventilation | 378 | 649 | 30 | 1847 | 38 | 2.14 | 4.81 | 302 | 565 | 26 | 1477 | 37 | 3.08 | 4.40 |
| **S2: With a fixed external shading device** | | | | | | | | | | | | | | |
| - without natural reinforced ventilation | 504 | 858 | 39 | 2079 | 36 | 0.67 | 1.03 | 764 | 1232 | 56 | 3544 | 38 | 0.86 | 1.03 |
| - with night natural reinforced ventilation | 124 | 308 | 14 | 486 | 34 | 1.40 | 4.12 | 234 | 516 | 24 | 1011 | 36 | 1.69 | 4.52 |
| - with day and night natural reinforced ventilation | 77 | 213 | 10 | 290 | 32 | 1.71 | 4.31 | 131 | 267 | 12 | 478 | 33 | 2.50 | 4.50 |
| **S3: With a moveable external shading device** | | | | | | | | | | | | | | |
| - without natural reinforced ventilation | 95 | 422 | 19 | 376 | 31 | 0.67 | 1.02 | 378 | 732 | 33 | 1257 | 33 | 0.86 | 1.02 |
| - with night natural reinforced ventilation | 7 | 82 | 4 | 34 | 29 | 1.21 | 4.13 | 66 | 255 | 12 | 242 | 31 | 1.55 | 4.33 |
| - with day and night natural reinforced ventilation | 3 | 47 | 2 | 17 | 29 | 1.43 | 4.17 | 24 | 147 | 7 | 101 | 30 | 2.16 | 4.24 |

OP/EP means ratio "overheating period" over "exploitation period". ODH means overheating degree-hours. * The ACH corresponds to the overall air change rate taking into account the air infiltration, the mechanical ventilation and the natural ventilation.

As expected, Table 6 shows that the high occupancy scenario corresponds to the worst-case scenario in all aspects. Moreover, it demonstrates that overheating risk is a serious problem if no appropriate measures are taken, even for the low occupancy scenario. By a combination of performant external shading with natural ventilation strategy, the overheating period can be limited to below 200 h, therefore, less than 10% of the exploitation period (218 h), which is the criterion imposed by Luxembourgish regulation. This is valid for both the "AAA energy class" and the slightly less insulated "building permit application" module variants. DTS showed that for all sub-variant configurations, most of the overheating hours occur when the outside temperature is lower than the room temperature. This means that the heat transfer goes from the interior to the exterior over the majority of the overheating hours. Since a better insulation performance corresponds to lower heat transfer, the higher insulation standard of the AAA energy class presents a noticeable disadvantage in terms of overheating hours as well as of overheating degree-hours. Furthermore, for both module variants, the overheating period of over 28 °C is not negligible, although the maximum room temperature reached is not relatively too high (31 °C). Hence, thermal comfort could be optimized, for instance with reference to the DIN 1946-2 standard [52], as shown in Figure 11.

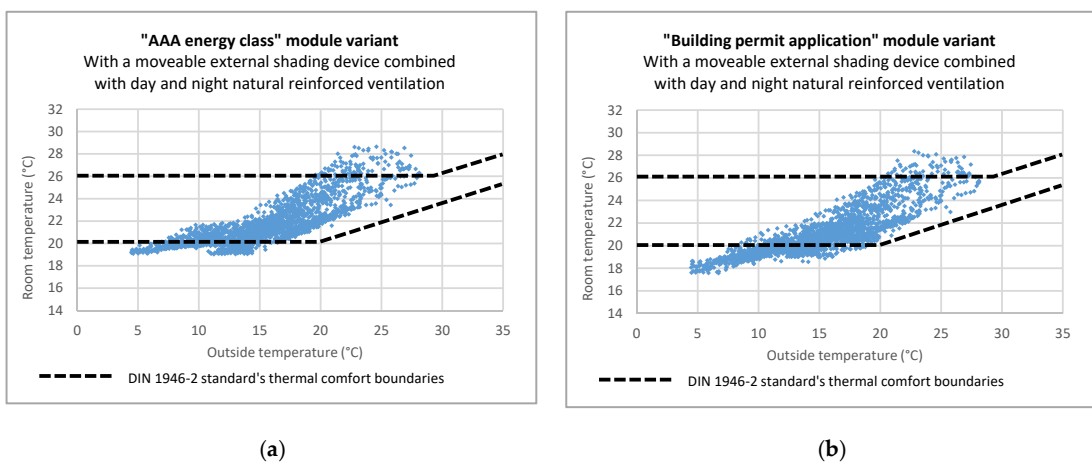

(**a**)  (**b**)

**Figure 11.** Room temperatures versus outside temperatures: (**a**) "AAA energy class" module variant; (**b**) "Building permit application" module variant.

As a comparison, DTS showed that the overheating period of the "AAA energy class" fictive module version can be limited to 65 h (3% of the exploitation period) and the maximum room temperature to 28 °C, which is even lower than the maximum outside temperature (30 °C). Therefore, the increasing of the thermal mass on the module has fundamentally improved thermal comfort as shown in Figure 12.

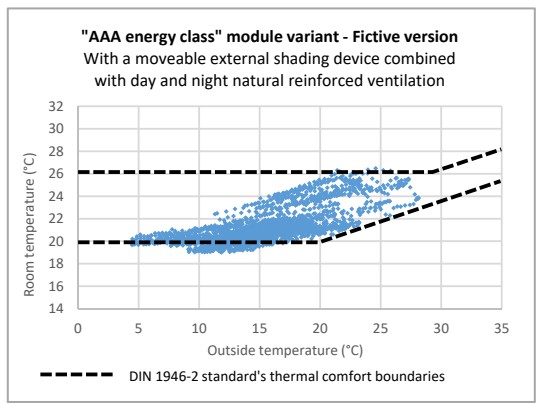

**Figure 12.** Room temperatures versus outside temperatures in the fictive module version.

## 6. Conclusions

A summertime overheating risk assessment on two module variants of the Slab building was carried out through the realization of DTS; the first variant fulfills the requirements for "AAA energy class" and the second one for "building permit application." Although the current Luxembourgish regulation does not yet cover the Slab building typology, it was used as a basis for this assessment, whereby the worst-case scenarios were taken as reference. These include the worst-case implementation site (lowest altitude in Luxembourg), the worst-case orientation (window facing south-west), the worst-case height (module located on the lowest floor of the Slab building) and the worst-case sun shading. Moreover, a low factor of window opening (tilt opening with an opening factor of 0.25 and 44% of operable part reported to the window surface) was assumed to ensure a comfortable ventilation for the high and the low occupancy. Three main sub-variants involving different external shading device settings and different natural reinforced ventilation strategies were considered. DTS realized on TRNSYS software showed that if no natural ventilation is applied, the effect of preventing overheating by using a moveable external shading device based on solar radiation is less significant in the "AAA energy class" module variant than in the "building permit application" module variant. Furthermore, the adoption of a good natural ventilation strategy tremendously reduced the overheating period, even without the use of a shading device. This implies that efficient natural ventilation has a preponderant effect in mitigating overheating compared with the use of a performant shading device. DTS results revealed that the overheating period for which the room temperature exceeds 26 °C can be limited to below 10% of the exploitation period, and this is valid for the two module variants. This means that the criteria for overheating defined by Luxembourg regulation are not fulfilled for both variants, particularly thanks to a moveable automated external shading device based on solar radiation, in combination with an automated day and night reinforced natural ventilation, which provides an overall ACH of up to 4.40 h$^{-1}$ with a mean value of 2.69 h$^{-1}$. This is quite expectable since the period for which the exterior temperature exceeds 26 °C for the climate of Luxembourg represents only 4% of the exploitation period. However, the overheating period of over 28 °C is not negligible although the maximum room temperature reached is not relatively too high (31 °C). Hence, thermal comfort can be optimized by increasing the thermal mass of the module walls, in this instance, with 5 cm of lightweight concrete. Since this is a simulation study, it has its limitations. Firstly, it focuses on thermal comfort assessment and does not provide an appreciation of visual comfort. Secondly, TRNSYS weather data files were updated in 2004 and are subsequently obsolete; moreover, future temperatures are expected to be higher. Lastly, the eventual impact of the shading device on the openings, and thus on the air exchange rate, was not taken into account. To conclude, Luxembourg regulation on energy efficiency of residential buildings needs some amendments, in a way that flexible plug-in constructions would not be severely penalized.

**Author Contributions:** Conceptualization, M.R. and F.S.; methodology, M.R. and F.S.; software, TRNSYS and SketchUp; validation, F.S.; writing—original draft preparation, M.R.; writing—review and editing, F.S. and D.W.; supervision, F.S.; project administration, D.W.; funding acquisition, D.W. All authors have read and agreed to the published version of the manuscript.

**Funding:** This research is in the framework of the ECON4SD (Eco-Construction for Sustainable Development) project, supported by the Luxembourgish EU programme "Investissement pour la croissance et l'emploi"—European Regional Development Fund (2014–2020) (Grant agreement: 2017-02-015-15).

**Acknowledgments:** The authors would like to express the gratitude to the whole team of the ECON4SD project, particularly to Marielle Ferreira Silva for providing the drawings and the 3D model of the Slab building.

**Conflicts of Interest:** The authors declare no conflict of interest.

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
