# Peer review of "Summertime Overheating Risk Assessment of a Flexible Plug-In Modular Unit in Luxembourg"

_sustainability, doi:10.3390/su12208474_

Round 1

Reviewer 1 Report

The core content of the paper is an accurate energy assessment of a prefabricated building systems, consisting in a set of living modules arrayed on the floors of a concrete structure acting as a support for the modules. Since the system is designed to be deployed in different locations and to face different orientations, the simulation properly considers of different sub-variants provided by the systems to meet these requirements

If it could clearly focus on just this goal, the paper would provide an interesting case study of thermal assessment carried according the regulation and adopting advances methodology.

This will need a more detailed description of the designed building system, better explaining the assembling rules and geometries (by providing schemes, cross sections, etc.), as well as better clarifying how the modules are produced, carried and put in operation. By the way: a 3 meter large module exceed the maximum width allowed in EU for a standard road transport, which is 2,5 meter (see: COUNCIL DIRECTIVE 96/53/E and subsequent amendments)

By adopting this approach, the Authors can avoid the discussion about how and why the plug-in flexible modular units have been adopted, simply assuming this option as a design brief decision relating to the recognized advantages of the off-site construction building systems, which are largely documented in the literature (as in Lines 64 to 80 of the current text)

On the contrary, if the off-site building system is part of the topics the paper discusses, this needs a deeper and more robust study on the wide available literature

On the contrary, if the modular off site building system is part of the topics the paper discusses, it needs a deeper and more robust study, referring to the wide available literature.

In particular, at least an overview must be provided on the different possible technical solutions, materials, module size by which the volumetric units can be made, as a reference framework useful to explain and justify the options adopted within the project.

This classification can also provide some critical element in order to correctly interpret the literature references regarding the thermal well being assured by the module (Lines 80 to 84 and 14 to 154), which strictly depends on the specific module technology, while it cannot be generically awarded to the prefabricated modular system in itself

Escaping a systematic classification of this large family of solutions leads in further incorrect (or at least unjustified) assertions, such as "the Buildings based on plug-in concept are barely referenced in literature" (Line 155) which must be explained by providing a stronger definition of "plug in concept".

The fact that " Regarding plug-in or flexible modular units, no studies on thermal comfort assessment have been found in literature (lines 134 to 135) must induce the Authors to questionning about the effectiveness of the query performed and the consistency of the notion of "plug-in or flexible modular units"

Reviewer 2 Report

In this paper, when plug-in type modular is applied to Luxembourg, the overheating phenomenon in summer was analyzed and the effect of measures to reduce the overheating phenomenon was analyzed.

I present the following review opinions.

The reader needs to understand the Luxembourg climate. Based on the meteorological data file, it is necessary to briefly represent Luxembourg's summer climate (temperature, humidity, wind direction, wind speed, solar radiation, etc.) in Section 4.3.1 as a graph.

Is the depth of the sunshade shown in Figure 3 an optimized value for insolation control? Indicate the basis for the depth value.

I have a question about Table 6. The S1 OP of “AAA energy class”, which appears to have enhanced insulation, is 2,132. On the other hand, the S1 OP of “Building permit application” with low insulation performance is 1,405. Shouldn't the S1 OP of “AAA energy class” with good insulation performance show a lower value?

Did you calculate the wind pressure acting on the exterior wall of the building by using the wind speed and wind direction data included in the weather data during ventilation simulation? I think this is an important process when calculating natural ventilation based on weather data.

Looking at the methodology of the study, it seems that natural ventilation and mechanical ventilation were applied. Reading the text, the maximum ventilation frequency seems to be set to 1 ACH. If this is correct,

Natural and mechanical ventilation can be defined as hybrid ventilation. In the case of HV, overheating can be prevented through a control strategy and the ACH number of ventilation can be significantly increased for indoor comfort. If overheating was prevented through HV control strategy in this study, present the results for the ACH number of ventilation.

The conclusion needs to provide more specific results. Qualitatively indicate the effect of preventing overheating by blocking the sun. In addition, qualitatively indicate the effect of overheating prevention by ventilation control strategy. Since this is a simulation study, it is necessary to describe the limitations of the study.

Line 102 :  ‘figure 1Error! Reference source not found.’ Should be fixed.

Overall, it is necessary to increase the numerical text size of the figures.

Round 2

Reviewer 1 Report

Since a strength of the modular construction is to be easily refitted, relocated and refurbished (as rightly reported in lines 77 and 78), a short description could be added explaining how the project met this requirements.

Author Response

The section 3 has been reworked to provide a short description explaining how the modules can be easily refitted, relocated and refurbished.

The modifications are from Line 192 to 198.

Reviewer 2 Report

Thank you again for submitting the author's paper to 'sustainability'.

The author's responses are satisfactory to some of the comments I raised during the last review process. And I think the paper has been faithfully revised.

I think that verification of the results is also important in dynamic thermal analysis simulation-based studies. Although TRNSYS is recognized for its accuracy worldwide and is an excellent analysis tool that is applied to many studies, simulation input conditions are very diverse and complex, and the results can vary depending on the skill level of the simulator. 

Therefore, although it is the author's choice, I recommend that you go through a verification process through comparison between the actual data and the simulation results as much as possible.

Finally, the background color in Figs 6, 7 and 8 should be changed to white.
